# Physiological and Metabolic Traits Linked to Kiwifruit Quality

**Vaia Styliani Titeli [1], Michail Michailidis [1,*], Georgia Tanou [2,3] and Athanassios Molassiotis [1]**

[1] Laboratory of Pomology, Department of Horticulture, Aristotle University of Thessaloniki, 57001 Thermi, Greece; titelivg@agro.auth.gr (V.S.T.); amolasio@agro.auth.gr (A.M.)

[2] Institute of Soil and Water Resources, Hellenic Agricultural Organisation-DIMITRA (ELGO-DIMITRA), 57001 Thermi, Greece; gtanou@swri.gr

[3] Joint Laboratory of Horticulture, Hellenic Agricultural Organisation-DIMITRA (ELGO-DIMITRA), 57001 Thermi, Greece

* Correspondence: msmichai@agro.auth.gr

**Abstract:** The assessment of fruit quality traits is a key factor in increasing consumer acceptance of kiwifruit. Here, an experiment was performed to evaluate the relationship between dry matter (DM) and soluble solids concentration (SSC), evaluated by both destructive (D) and non-destructive (ND) approaches, with acidity content and sensory evaluation, particularly taste, in fully ripened 'Hayward' kiwifruits from 20 orchards. Nutrient content and metabolomic analysis were also performed in ripened kiwifruit tissues (pericarp, placenta, and seeds) from four selected orchards of kiwifruits of high taste scores (HTS) and four orchards of low taste scores (LTS). The results suggest that ND-DM measurement positively correlated with D-DM and may serve as an indicator of kiwifruit taste. Moreover, the taste of kiwifruit was affected by both SSC and acidity. Based on the nutrient content of the pericarp and the primary metabolites of the pericarp and placenta, a clear separation was observed between kiwifruits with HTS and those with LTS, while no differences were found in seed samples. Metabolites such as fructose, maltose, mannobiose, tagatose, and citrate were accumulated in kiwifruits with a strong taste in the pericarp, whereas others, such as serine in the pericarp and placenta, have a negative impact on taste. The current study contributes to a greater understanding of the influence of dry matter, ripening characteristics, primary metabolites, and nutrient content on the taste of kiwifruits.

**Keywords:** dry matter; metabolomics; sensory evaluation; taste; elements

## 1. Introduction

The dietary and nutritional value of kiwifruit is highly appreciated by consumers [1,2]. Fruit ripening, as a process, is signaled by changes in the sensory components, including taste (e.g., increased sweetness), visual appearance (internal color change), texture (softening of the flesh), and aroma (formation of volatile organic compounds, VOCs) [3]. Aroma and taste primarily develop during the final stages of ripening in climacteric fruit such as kiwifruit, where they are associated with autocatalytic ethylene production [4]. Consequently, flavor, texture, aroma, nutritional elements, and expiration life are the primary characteristics that influence consumers' preferences and determine a product's quality. Kiwifruit flavor is the most important quality indicator and characterized by balance of sweetness and acidity with a fine blend of volatile aromatic compounds [5], illustrating how these specific characteristics are affected by the ratio between sugars and acids [6,7].

The soluble solid concentration (SSC) is recognized as the primary ripening criterion for kiwifruits, which are harvested when the SSC content exceeds 6.25% Brix. SSC content depends on the accumulation of sugars in the form of starch/dry matter during fruit ripening [7], and there is a direct correlation between SSC content and sweetness in flavor [8]. Dry matter is also another crucial quality indicator for kiwifruit; it refers to the non-water components of the fruit and, whether there are photosynthesizing leaves, it increases

as the fruit ripens [9]. According to Feng [9], the anticipated concentration of SSC in ripe fruit corresponds to the percentage of dried weight at harvest, reduced by three percent. Dry weight estimation is an accepted indicator of fruit ripening and a predictor of SSC in New Zealand [10], allowing producers to obtain higher prices [11]. Crisosto [8] noted the reliability of dry matter as an indicator of quality, but not ripening, of kiwifruit, which confirms the increased consumer acceptability of high dry weight kiwifruits [5]. Liao [12] found that cultivation practices such as kiwifruit canopy formation and summer pruning influence the fruit's dry matter content. In addition, the final estimation of all these fruit quality traits that may affect flavor is being studied through sensory evaluation. Sensory analysis is a technique that provides valuable information that food processing industries use to evaluate the quality of their products [13]. To be more representative, comparisons and correlations between outcomes of sensory analysis must be quantified and rated [14]. Consequently, the correlation between the sensory evaluation's results and objective measurements provides the highest degree of reliability when calculating fruit quality index [15].

The purpose of this study is to (i) establish a relationship between destructive and non-destructive methods for evaluating the quality of fully ripened kiwifruits, and (ii) reveal possible kiwifruit quality indicators, utilizing sensory evaluation and particularly kiwifruits' taste from different orchards and from separated fruit tissues (pericarp, placenta, and seeds) and their association with nutrient content and metabolites abundance.

## 2. Materials and Methods

### 2.1. Experiment and Fruit Sampling

Kiwifruits (*Actinidia chinensis* var. *deliciosa* (A. Chev.) A. Chev. 'Hayward') were harvested from 20 productive (all above 5 years old) commercial orchards (codes 1–20) that are located in the Nestos area (East Macedonia, North Greece; Figure S1). All orchards were subjected to standard cultural practices for sustainable kiwifruit production. During the commercial harvest period, 200 fruits per orchard, with an average soluble solid concentration of SSC = 6.25% Brix and fruit weight of approximately 100 g, were collected. Fruit were transferred to the laboratory of Pomology at Aristotle University and, following postharvest storage at 0 °C (RH = 95%) for three weeks, were then maintained at 20 °C (RH = 70%) until ripened (pericarp firmness lower than 10 Newtons). In the fully ripe stage of kiwifruits, both non-destructive and destructive approaches were used to determine dry matter and SSC; additionally, acidity as well as sensory evaluation were performed. In parallel, the kiwifruits were divided into three distinct tissues: pericarp, placenta, and seeds, followed by sampling with liquid nitrogen and storage at −80 °C for further analysis.

### 2.2. Fruit Quality Traits

2.2.1. Non-Destructive Estimation of the Dry Matter and Soluble Solid Concentration

The non-destructive determination of kiwifruit dry matter (%) and SSC (%) were carried out in 30 kiwifruits per orchard at the fully ripe stage using an F-750 device (NIR spectroscopy-based technology, Produce Quality Meter, Felix Instruments, Camas, WA, USA) with the software of F-751 Kiwi Quality Meter. Initially, the F-750 device was calibrated with 20 fruits at 20 °C (RH = 70%). The fruit was placed in the special holder and the reading was taken according to the manufacturer's instructions. Thereafter, non-destructive determination of dry matter (%) and SSC (%) were conducted in the kiwifruits from 20 orchards.

2.2.2. Destructive Determination of Dry Matter, Soluble Solid Concentration, and Acidity

Dry matter content was determined in a transverse slice (1 cm) from the equatorial region of each fruit, when dried in an oven (65 °C, 48 h), and expressed as a percentage (%) of dry weight against fresh slice weight. Soluble solid concentration (SSC, %) and titratable acidity (TA) were determined in juice from three biological replicates of 10 kiwifruits per orchard. SSC measurement was performed using a digital refractometer (Atago PR-101,

Atago Co., Ltd., Tokyo, Japan) while TA was performed by potentiometric titration with 0.01 NaOH up to pH 8.2 and was expressed in citric acid (%), as previously described [16].

### 2.3. Sensory Evaluation Analysis

Panelists evaluated the ripe kiwifruits of cultivar 'Hayward' under normal light, temperature, and relative humidity conditions, as previously described [17]. The sensory descriptors that were determined based on intensity scale were appearance (1 = abnormal fruit with defects and 9 = oval shape without any defect), internal color (pericarp green intensity; 1 = white; R, G, B = 255, 255, 255; 5 = light green; R, G, B = 0, 180, 0; 9 = dark green; R, G, B = 0, 120, 0), texture/firmness (1 (4 N) to 9 (10 N)) from low to high firmness based on chewing of pericarp and placenta) [18], aftertaste intensity (1 = absent and 9 = very strong), taste intensity (1 = without taste; taste of mineral water and 9 = high; very tasteful), as previously described with some modifications [19]. Each sample was labeled with a random four-digit code, containing five whole kiwifruits of each orchard, and then transverse slices (1 cm) of each fruit were presented on white polystyrene plates balanced in a randomized order across the nine judges/panelists. A training session was organized with the purpose of acquainting the panelists with the sensory evaluation approach, attribute definitions, and reference standards. Additional information about the tested parameters is provided in Table S1.

### 2.4. Mineral Element Analysis

Determination of nutrients (K, P, Ca, Mg, Na, Zn, Fe, Mn, Cu) was performed in the pericarp of ripe kiwifruits (dry samples) by the inductively coupled plasma optical emission spectrometry (ICP–OES) system (Perkin Elmer Optima 2100DV, PerkinElmer Inc., Waltham, MA, USA). A batch of five kiwifruits in three biological replicates per orchard was used, and extraction was carried out on ash dissolution in 6 N HCl, after sample incineration at 550 °C for 6 h. The Kjeldahl method was used to perform the analysis of nitrogen (N) with a Vapodest 50 s system (Gerhardt, Königswinter, Germany) [20].

### 2.5. Primary Polar Metabolites Analysis

Primary polar metabolites extraction of ripe kiwifruits in each tissue (pericarp, placenta, seeds) and derivatization processes were employed, as previously described [16], with slight modifications. Frozen grinding tissue (0.5 g) from eight selected orchards (four orchards that received a high score based on taste and four orchards that received a low score based on taste) were used. Samples were extracted with 1.4 mL of methanol (100%, pre-cooled at $-20$ °C), and 0.1 mL adonitol (1 mg mL$^{-1}$) was added as an internal quantitative standard, and then incubated for 10 min (70 °C). In the supernatant, 0.75 mL of chloroform (100%, pre-cooled at $-20$ °C) and 1.5 mL dH$_2$O (100%, pre-cooled at 4 °C) was added and then centrifugated (2200× $g$, 4 °C, 10 min). An aliquot of 0.15 mL of the supernatant was transferred into a vial glass and placed to dry in a desiccator under vacuum. The residues were redissolved in 0.04 mL methoxyamine hydrochloride (20 mg mL$^{-1}$) and then in 0.07 mL N–methyl–N–(trimethylsilyl) trifluoroacetamide reagent (MSTFA) for 120 min and 30 min at 37 °C, respectively. The GC–MS analysis was carried out with a Perkin Elmer Clarus™ SQ 8 (Waltham, MA, USA), as previously described in detail [16]. Compounds were determined using standards or NIST11 database or GOLM metabolome database [21]. The metabolites were expressed as the relative abundance of adonitol and are provided in Table S2.

### 2.6. Statistical Processing and Analysis

The statistical analysis of all quality traits, sensory attributes, and mineral elements was conducted using SPSS (SPSS v27.0., Chicago, IL, USA) by one-way ANOVA or by multivariate analysis of variance (MANOVA) [22]. Metabolomic analysis was performed by analysis of variance (ANOVA) between high- and low-taste-score kiwifruits. Mean values were compared based on the least significant difference (LSD) or Student's *t*-test; $p \leq 0.05$.

Pearson correlation analysis was conducted using SPSS whereas principal components analysis (PCA) and hierarchical clustering along with a heatmap were employed using ClustVis software 2.0 [23].

## 3. Results

### 3.1. Non-Destructive Application to Define Fruit Quality

To examine the internal kiwifruit quality at the fully ripe stage from the 20 orchards, both non-destructive and destructive approaches were used to determine dry matter (DM) and soluble solid concentration (SSC), as well as the acidity content and the sensory attributes (Figure 1).

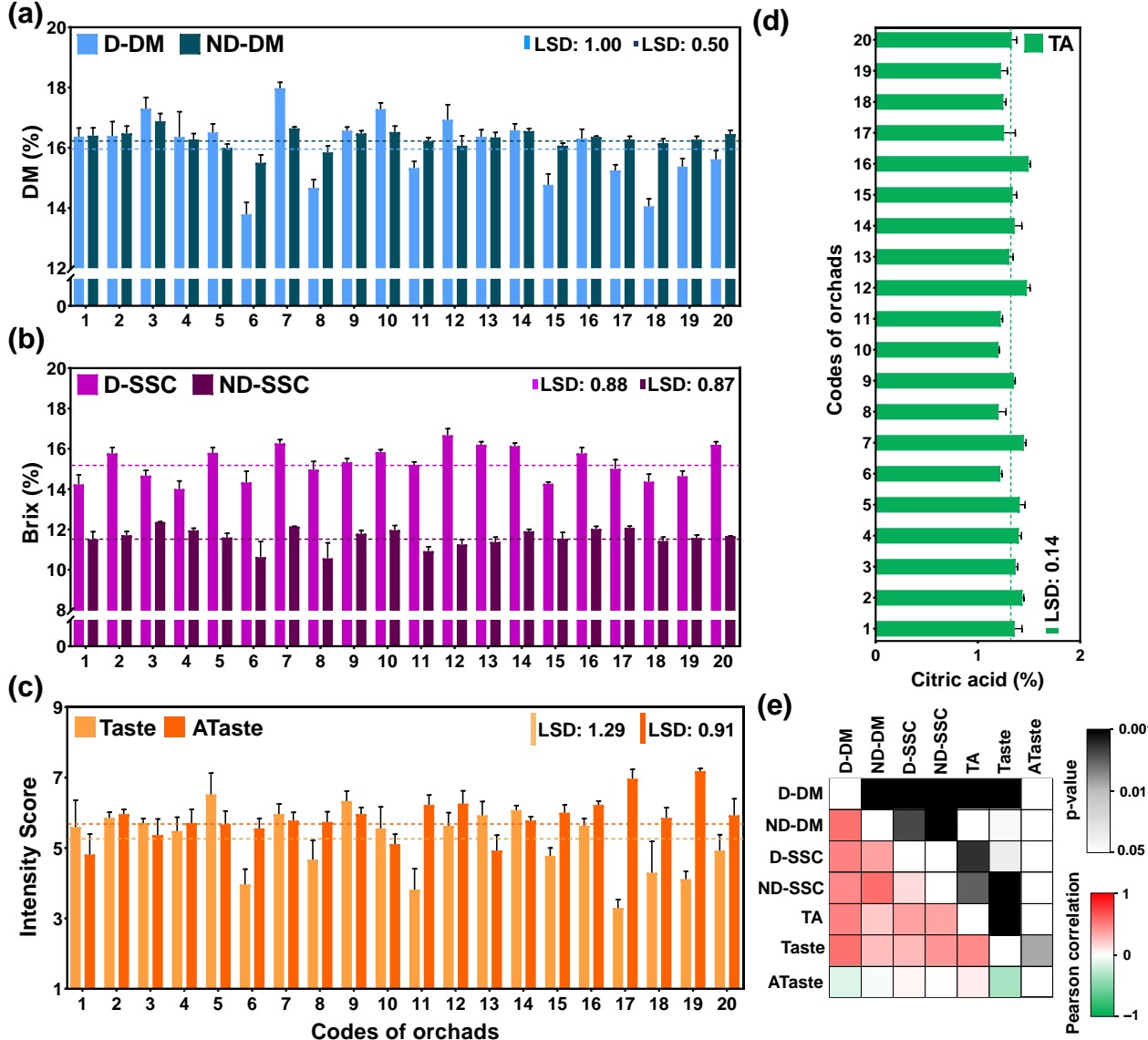

**Figure 1.** Quality traits of fully ripe fruits from 20 kiwifruit orchards: (**a**) dry matter (%) of fruit using destructive (D-DM) and non-destructive (ND-DM) approach, (**b**) soluble solids concentration (% Brix) of fruit using destructive (D-SSC) and non-destructive (ND-SSC) methods, (**c**) sensory evaluation of kiwifruit base on intensity score ranged from 1 (without taste) to 9 (very strong taste/aftertaste) of taste and aftertaste (ATaste), (**d**) titratable acidity (TA), and (**e**) Pearson correlation of seven variables depicted with a heatmap (red indicates positive whereas green indicates negative correlation). The dotted line indicates the grand mean of its variable. The means of its quality and sensory trait were compared based on least significant difference (LSD) $p \leq 0.05$.

The average dry matter (DM) content at the 20 orchards was 16% by the destructive (D) and 16.3% by the non-destructive (ND) approach. The higher DM was recorded at orchard codes 3 and 7 and the lower DM was recorded at 6 and 8 orchard codes both in D and ND approaches (Figure 1a). The average soluble solid concentration (SSC) at the 20 orchards was 15.3% and 11.6% by D and ND approaches, respectively. It is worth noting that ND-SSC was significantly underestimated with respect to D-SSC. Nevertheless, a higher SSC was recorded in both D and ND approaches at orchard code 7. Additionally, high SSC was observed at orchard codes 12 for D and 3 for ND, whereas the lower SSC was at orchard codes 4 and 1 for D and orchard codes 6 and 8 for ND (Figure 1b). We should also highlight that orchard codes 3 and 7 received high values of DM and SSC, while orchards 6 and 8 received low values of DM and SSC.

Sensory evaluation revealed an acceptable intensity score ($\geq$5) for most of the orchards analyzed for descriptors such as appearance, internal color, and texture/firmness intensity in fully ripe kiwifruits (Table S1). We assumed that taste and aftertaste are two crucial descriptors in kiwifruit sensory evaluation. Thus, we were focused on these two descriptors; the average taste score was 5.2 and the average aftertaste score was 5.9 among the 20 orchards. The four higher scores in taste were observed at 7, 14, 9, and 5 codes of orchards whereas the four lower scores were observed at 17, 11, 6, and 19 codes of orchards (Figure 1c). In contrast, aftertaste (Ataste) results were recorded at a higher intensity in low-taste-score orchards, such as 17 and 19, indicating a negative relationship between taste and aftertaste in kiwifruits. Finally, the average value of acidity (TA) content at the ripe kiwifruits of 20 orchards was 1.3%, with the higher TA being recorded at orchard codes 12 and 16 and the lower TA at orchard codes 10 and 8 (Figure 1d).

To uncover possible associations among the seven variables tested, a Pearson correlation analysis was performed. This examination revealed positive correlations between destructive and non-destructive approaches among DM, SSC, TA, and taste, whereas a negative correlation was observed between taste and Ataste intensity (Figure 1e). Hence, considering the above-mentioned correlations and to evaluate the factors affecting the taste in ripe kiwifruit, we chose the four orchards with the highest scores (7, 14, 9, and 5 codes of orchards) and four orchards with the lowest scores (17, 11, 6, and 19 codes of orchards) according to their taste assessment for further analysis.

### 3.2. Taste-Dependent Quality Traits and Mineral Content

To estimate the impact of taste in ripe kiwifruit, the four orchards that received a high taste score (HTS) were compared with the four orchards that received a low taste score (LTS). Physiological analysis revealed that the HTS orchards exhibited higher values of dry matter, SSC, and acidity than the LTS orchards, whereas LTS orchards had higher Ataste intensity than HTS (Figure 2a). In contrast to the physiological data, the mineral content in the pericarp of ripe kiwifruit displayed a decrease in HTS compared with LTS orchards, concerning the content of nitrogen, phosphorus, potassium, iron, and zinc (Figure 2b). However, the content of calcium, magnesium, sodium, copper, and manganese was similar between the ripe kiwifruits that received high and low taste scores (Figure 2b).

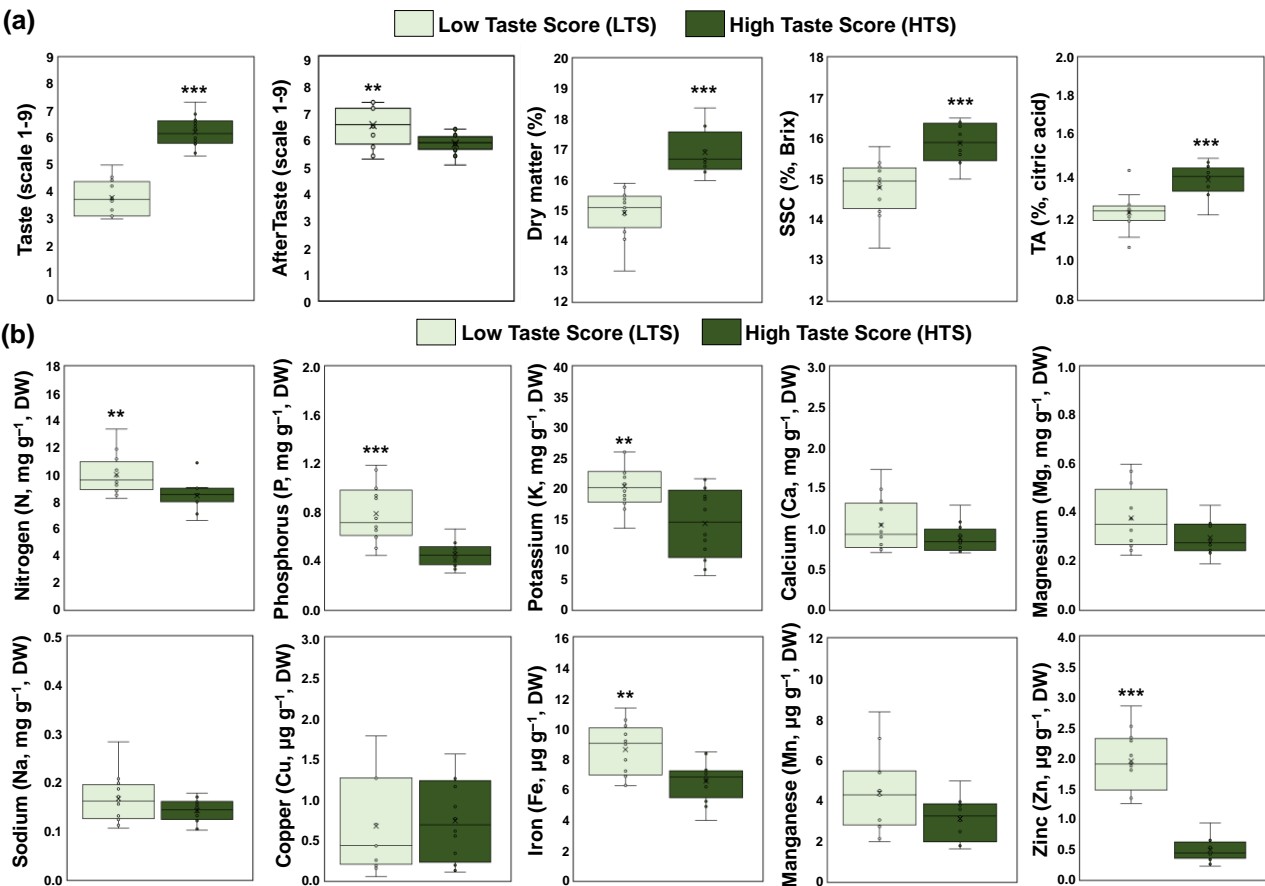

**Figure 2.** (**a**) Sensory evaluation (taste and aftertaste intensity) and physiological traits (dry matter, SSC, and TA) of fully ripe kiwifruits with high taste scores (HTS, dark green) and low taste scores (LTS, light green), and (**b**) mineral content in the pericarp of ripe kiwifruits with high taste scores (HTS, dark green) and low taste scores (LTS, light green). Each box plot was constructed by 12 replicates (four codes of orchards × three biological replicates). Means were compared based on Student's *t*-test; * $p \le 0.05$, ** $p \le 0.01$, *** $p \le 0.001$.

### 3.3. Taste-Associated Primary Metabolites in the Various Kiwifruit Tissues

To identify the contribution of primary metabolites in the taste of kiwifruit at the fully ripe stage, a primary polar metabolomic analysis in three kiwifruit tissues (pericarp, placenta, and seeds) was performed. The orchards that received a high taste score (HTS) were compared with the orchards of LTS in each tissue separately. In pericarp tissue, 42 metabolites were identified belonging to five classes, namely sugars (16), acids (11), alcohols (6), amino acids (7), and other compounds (3). A clear separation was observed between HTS and LTS in the pericarp of ripe kiwifruits based on hierarchical cluster analysis, as depicted in Figure 3a. Pericarp metabolomic analysis exhibited a decrease of two amino acids (valine and serine), one alcohol (inositol), and one other compound (phosphoric acid) in the HTS kiwifruits (Figure 3a, Table S2). On the contrary, an increase in nine sugars (tagatose, fructose, glucose, talose, sucrose, turanose, lactulose, mannobiose, and maltose), three acids (oxalic acid, citric acid, and quininic acid), one alcohol (galactinol), and one other compound (aucubin) was detected in the HTS kiwifruits (Figure 3a, Table S2). Furthermore, we found an increase in sugars and acids as well as a reduction in amino acids and other compounds in the HTS kiwifruits (Figure 3b).

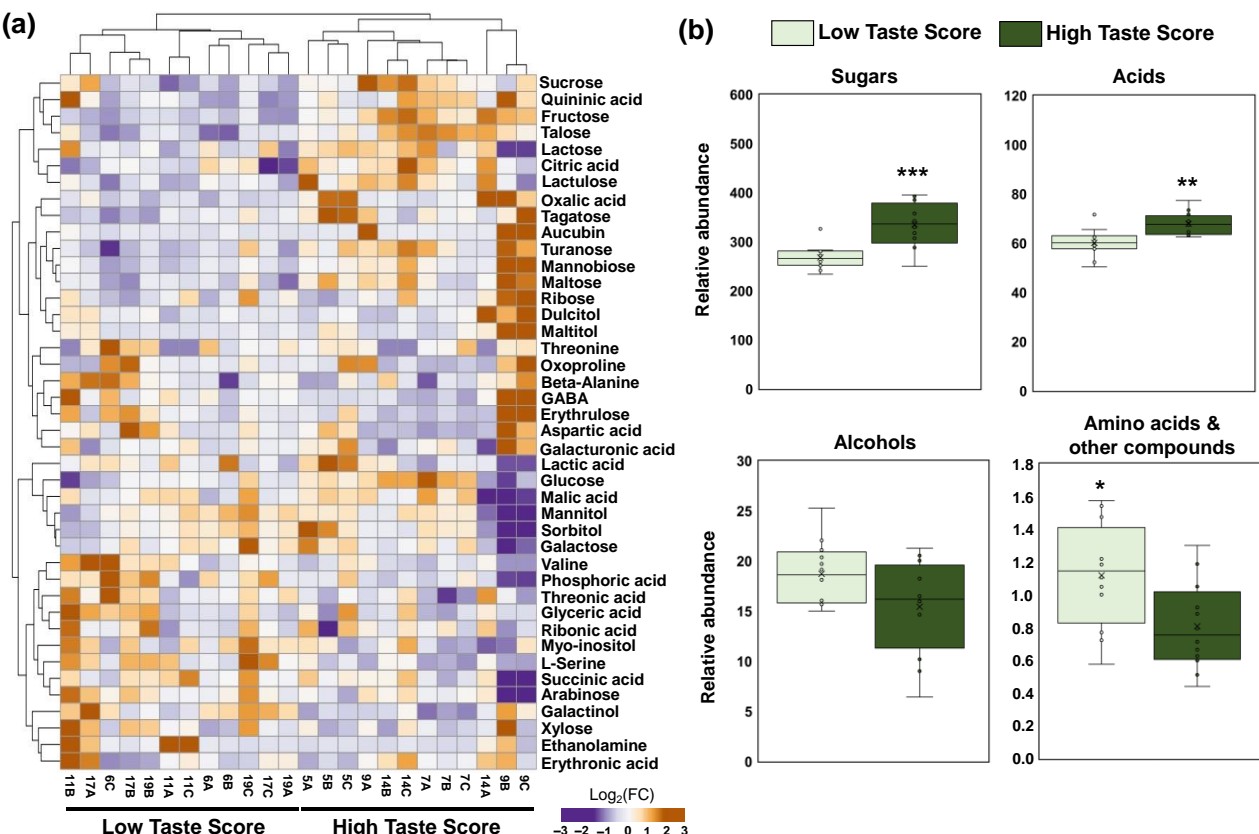

**Figure 3.** (**a**) Heatmap and hierarchical cluster analysis of the primary metabolites in the pericarp of the fully ripe kiwifruits with high taste scores (HTS; codes of orchards 5, 7, 9, 14) and low taste scores (LTS; codes of orchards 6, 11, 17, 19) using ClustVis software. The orange color indicates an increase in abundance; the purple color indicates a decrease in the abundance of each metabolite. (**b**) Metabolite classes in the pericarp of fully ripe kiwifruits with high taste scores (HTS, dark green) and low taste scores (LTS, light green). Each box plot was constructed by 12 replicates (four codes of orchards × three biological replicates). Means were compared based on Student's *t*-test; * $p \leq 0.05$, ** $p \leq 0.01$, *** $p \leq 0.001$. Metabolite data are provided in Table S2.

The current metabolomic analysis also identified 42 metabolites in the placenta tissue. In addition to this, 32 metabolites were detected in seeds tissue that correspond to five classes, namely sugars (16), acids (6), alcohols (6), amino acids (3), and other compounds (1). Based on hierarchical cluster analysis, a distinct separation was also noticed between HTS and LTS in the placenta of ripe kiwifruits but not in seeds, as illustrated in Figure 4. Metabolomic analysis in the placenta tissue disclosed a decrease in HTS of two sugars (xylose and galactose), two alcohols (sorbitol and galactinol), one acid (lactic acid), and three amino acids (serine, GABA, and oxoproline) (Figure 4a, Table S2).

An increase in metabolic abundance in the placenta of HTS kiwifruits was detected regarding six sugars (tagatose, turanose, lactulose, mannobiose, maltose, and lactose), one acid (erythronic acid), one alcohol (maltitol), and one other compound (aucubin) (Figure 4a, Table S2). In addition to this, seed metabolomic analysis showed a decrease in oxalic acid (acid) in HTS kiwifruits (Figure 4b, Table S2). In the seed of HTS kiwifruits, we also noticed an accumulation of three sugars (glucose, trehalose, and turanose), citric acid (acid), and aucubin (other compounds) (Figure 4b, Table S2).

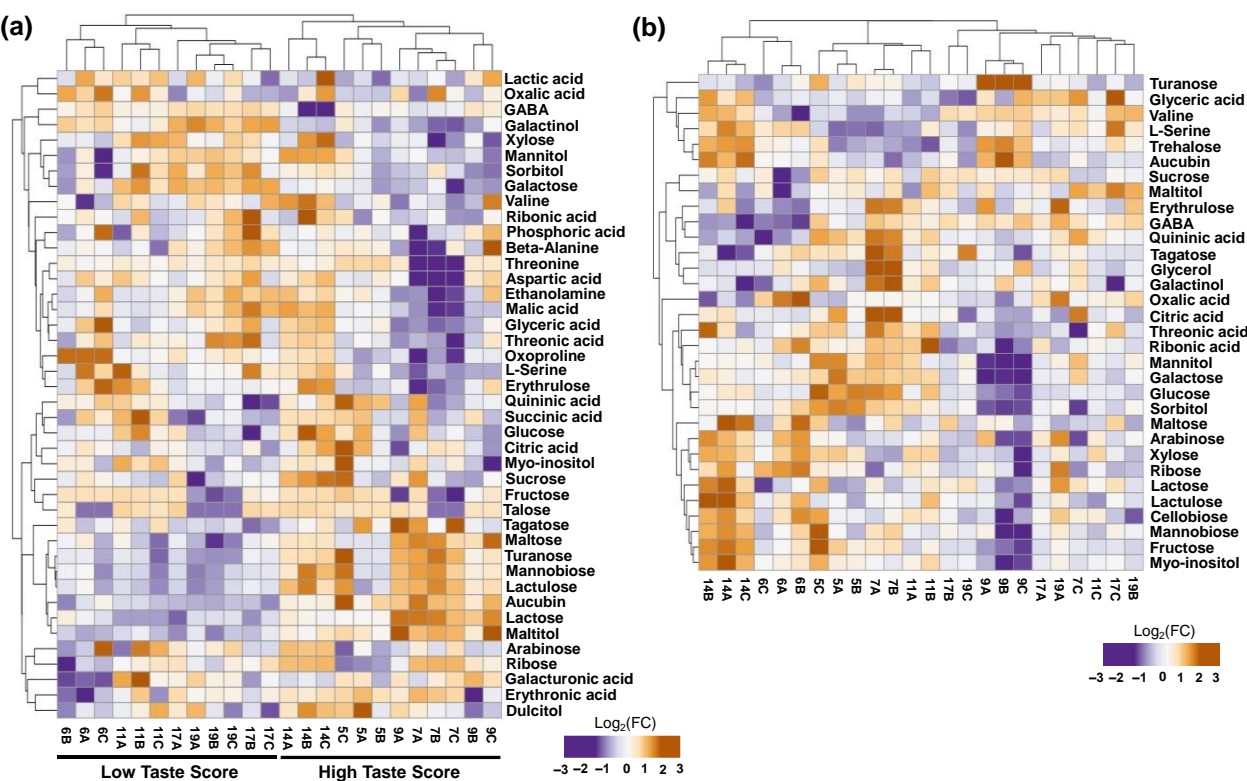

**Figure 4.** Heatmap and hierarchical cluster analysis of primary metabolites in the (**a**) placenta and (**b**) seeds of fully ripe kiwifruits with high taste scores (HTS; codes of orchards 5, 7, 9, 14) and low taste scores (LTS; codes of orchards 6, 11, 17, 19) using ClustVis software 2.0. The orange color indicates an increase in abundance; the purple color indicates a decrease in abundance. Metabolite data are provided in Table S2.

*3.4. Taste-Based Discrimination of Fully Ripe Kiwifruits*

To understand the effect of nutrients and metabolites on taste, a principal component analysis (PCA) was performed (Figure 5). Based on PCA of nutrients in the pericarp, we detected a clear separation between kiwifruits with high taste scores (HTS) and low taste scores (LTS) (Figure 5a). Similarly, a distinct separation was observed when PCA was performed in the metabolites of the pericarp and placenta but not in seeds between HTS kiwifruits and LTS ones (Figure 5b). It was also obvious that the discrimination of kiwifruits based on taste in the pericarp and placenta was mainly due to PC1, which explained the 20.7% variability of metabolites (Figure 5b). Focusing on metabolites that mainly participated in the construction of PC1, we noticed that six sugars (fructose, turanose, mannobiose, maltose, talose, and tagatose) and citric acid (the main acid in kiwifruits) received a positive score, above 0.6, indicating an increase in their abundance in the pericarp of HTS (Figure 5c). On the contrary, two amino acids (serine and valine) and phosphoric acid scored below −0.6, indicating an increase in their abundance in the pericarp of LTS (Figure 5c). In parallel, six sugars (lactose, turanose, maltose, lactulose, tagatose, and mannobiose), maltitol, and aucubin received scores above 0.6, implying an increase in their abundance in the placenta of HTS, whereas three amino acids (aspartic acid, GABA, and serine), two sugars (xylose, galactose), three acids (glyceric acid, threonic acid, and malic acid), galactinol, and ethanolamine scored below −0.6, indicating an increase in their abundance in the placenta of LTS kiwifruit (Figure 5c).

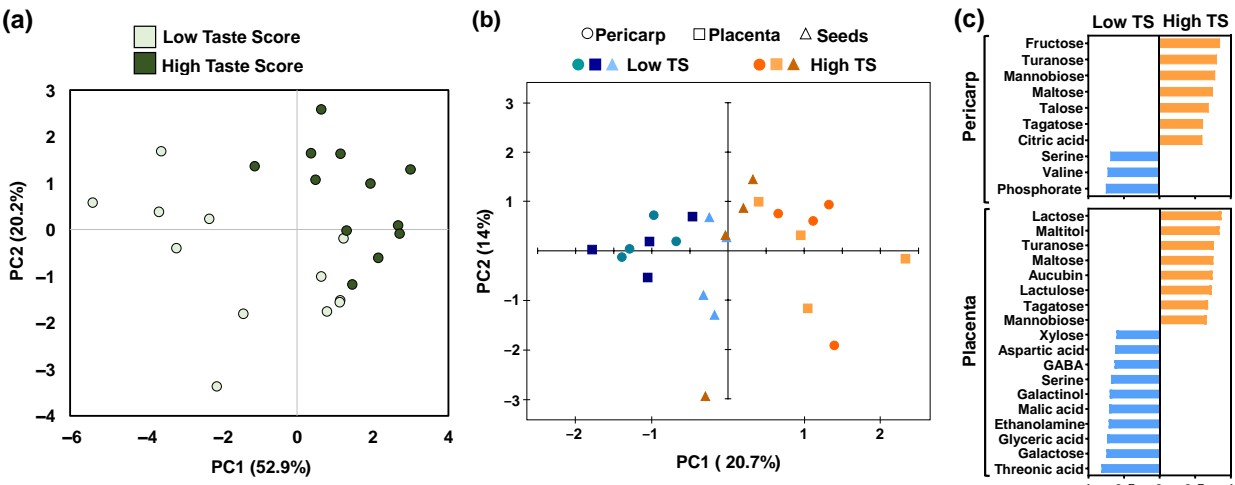

**Figure 5.** Principal component analysis (PCA) of the (**a**) minerals in fully ripe kiwifruits with high taste scores (HTS) and low taste scores (LTS), and (**b**) primary metabolites in the pericarp (cycle), placenta (square), and seeds (triangle) of fruit with high taste scores (HTS) and low taste scores (LTS). (**c**) Discrimination of LTS and HTS in the pericarp and placenta samples based on the metabolites using PCA models.

## 4. Discussion

It is well known that, during kiwifruit ripening, an increase in levels of SSC, ethylene, and volatiles, as well as a decrease in acids, starch content, and firmness, occur [5]. Flavor generation compounds are linked to both aroma (volatile compounds) and taste (sweet and acidic) in kiwifruit undergoing ripening. It is also well documented that soft kiwifruits, due to the activation of cell-wall-related enzymes such as polygalacturonase [24–26], are sweeter, whereas firmer kiwifruits are more acidic. Generally, values of firmness of approximately 8–10 Newtons make the kiwifruit ready to eat [27], while fruits with a slightly softer texture are more acceptable [18]. Therefore, our study focused on ripe kiwifruits, below 8 Newtons in the pericarp firmness, to determine quality traits such as SSC, DM, acidity, sensory attributes such as taste, etc. (Figure 1).

In the present study, considerable variation in fruit DM and SSC were evident among orchard samples of kiwifruit (Figure 1). Such variation in kiwifruit DM and SSC observed in this study was similar to that reported elsewhere [6]. For kiwifruit, the application of NIR-spectroscopy has been proposed as a tool for a quality metric and a harvest maturity metric [28]. It was also interesting that the non-destructive determination of DM and SSC using NIR-spectroscopy could develop strategies for appropriate postharvest handling of kiwifruits [2]. Especially, the SSC in kiwifruits should be higher than 6.2% Brix at harvest [29], while high DM (above 16%) was proposed as a quality indicator in kiwifruits [30]. It has been suggested that SSC should range between 10–14 °Brix in ready-to-eat kiwifruits [6]. In the current study, SSC ranged above these values from 14 to 16.7% Brix (Figure 1).

Since the DM of kiwifruit shows only minor change over the postharvest period [30], the measure of fruit DM at harvest may be used to predict the kiwifruit sensory potential following cold storage. In addition, the high level of dry matter in kiwifruits is linked to the acceptance of consumers with respect to the taste, since their sweetness increased [6,10,29,31–33]. In the present study, we also noticed that quality characteristics of kiwifruits (DM, SSC, and acidity) were positively associated with their taste scores (Figures 1 and 2a). As a result, DM content, SSC, and acidity can be considered as indicators related to other intrinsic quality attributes to improve the precision of the sorting process.

The nutrient content of the fruit may be contributed to the proportion of dry matter [5]. Nevertheless, there is still an extensive gap in our understanding concerning the

connection between nutrient homeostasis and dry matter of kiwifruit. In this regard, an interesting finding that emerged from this work is the fact that nutrient elements tend to elevate their content in the pericarp tissue of LTS kiwifruits (Figure 2b), suggesting that it can be possibly used for discrimination between HTS and LTS kiwifruits (Figure 5a). This connection between fruit nutrient concentration and DM needs further attention, particularly considering the fact that mineral elements have been previously analyzed to achieve fruit-part discrimination [34].

Central carbon metabolism, which in fruits involves the pathways of sucrose, starch, major organic acids, and respiration, provides energy and biosynthetic precursors to support fruit growth and, eventually, ripening [35]. It is also essential for fruit quality, as sweetness and sourness are conditioned by sugars and organic acids, respectively, which are major components in most fruits [35]. Focusing on taste via discrimination in high- and low-taste-score kiwifruits, we examined the effect of the primary metabolites in the different parts of kiwifruits, including pericarp, placenta, and seeds (Figures 3 and 4). The metabolic profile in kiwifruit pericarp displayed an enhancement of sugars and acids in kiwifruit that received a high taste score (Figure 3b). The interplay between SSC and acidity has been previously pointed out in kiwifruits for achieving high rates of consumer acceptance [5,6]. Moreover, no strong changes in the primary metabolites of seed samples were observed between the HTS and LTS groups of kiwifruit (Figure 4b). Recently, it was proposed that seeds' presence (depending on the pollen donor) in kiwifruits did not affect taste balance [36], reinforcing our observation.

Based on the high taste score (HTS), a clear discrimination in primary metabolites in both pericarp and placenta tissues was recorded (Figures 3–5), probably due to hydrolysis of starch to glucose during the postharvest period resulting in sugar accumulation in HTS kiwifruit [37]. It was documented that metabolites along with sensory properties influence taste in a variety of foods [38,39]. The discrimination between HTS and LTS pericarp samples may be related to the levels of nine metabolites (Figure 5c), among them, citric acid which is the main acid in cv. 'Hayward' [40,41] and fructose which is one of the main sugars in kiwifruits [16,42]. Hence, the elevated abundance of fructose and citric acid in HTS could be associated with the taste as well as with the observed increase in acids and sugars in HTS kiwifruits (Figure 3). It is also worth noting that other sugars such as turanose, maltose, tagatose, and mannobiose (these sugars in kiwifruits were found in low abundance [16,42]) were also responsible for the distinct patterns between HTS and LTS in the pericarp and placenta tissues (Figure 5c). Moreover, an amino acid, serine, increased in the placenta and pericarp of LTS fruit (Figure 5c), indicating that this metabolite may be associated with the low-taste feature in kiwifruit. Recently, serine has been positively related to quality features in guava fruits [43].

## 5. Conclusions

This study indicates that non-destructive methods for determining dried matter can be used as an internal indicator of kiwifruit taste. In addition, the taste of kiwifruit was influenced by both SSC and acidity and it was the most important sensory characteristic in terms of overall acceptability. There was also evidence of discrimination between kiwifruits with high and low taste scores based on the nutrient content and primary metabolites of the pericarp and placenta. Several metabolites (fructose, maltose, mannobiose, tagatose, and citrate) were elevated in kiwifruits with a strong taste, whereas others, such as serine, may have a negative impact on taste. This study revealed the significance of certain minerals and metabolites in taste discrimination and set a basis for future research on the quality of kiwifruit.

**Supplementary Materials:** The following are available online at https://www.mdpi.com/article/10.3390/horticulturae9080915/s1, Figure S1. Sampling area and location of kiwifruit orchards; Table S1. Sensory analysis of ripe kiwifruits; Table S2. Quantitative results of metabolites in ripe kiwifruit tissues.

**Author Contributions:** Conceptualization, M.M. and A.M.; data curation, M.M., V.S.T., G.T. and A.M.; formal analysis, M.M. and V.S.T.; investigation, V.S.T.; methodology, V.S.T.; project administration, A.M.; software, M.M.; supervision, A.M.; validation, M.M., G.T. and A.M.; visualization, M.M. and V.S.T.; writing—original draft, M.M. and V.S.T.; writing—review and editing, G.T. and A.M. All authors have read and agreed to the published version of the manuscript.

**Funding:** This work was funded by the Action "Investment Plans of Innovation" of the Operational Program "Eastern Macedonia—Thrace 2014-2020", which is co-funded by the European Regional Development Fund and Greece (Project code: AMTHR4-0040267).

**Data Availability Statement:** The data presented in this study are available in Supplementary Material.

**Acknowledgments:** We would like to thank Chrysanthi Polychroniadou and candidate/students Elpida Nasiopoulou, Christina Skodra, and Iasonas Zacharis for helping us during the sampling process and determination of kiwifruits' physiological quality traits.

**Conflicts of Interest:** The authors declare no conflict of interest.

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
