# Peer review of "Physiological and Metabolic Traits Linked to Kiwifruit Quality"

_horticulturae, doi:10.3390/horticulturae9080915_

Round 1

Reviewer 1 Report

The article has written very well, but there are few points that should be considered:

1- It is suggested to use keywords which have not mentioned in Title. (Different words in keywords than what is find in Title of the manuscript.

2- Please, pay attention to Paragraphing, especially in Introduction part, each paragraph should begin with new contents and words, and in the introduction some paragraphs can be together.

3- Materials and methods have written very well, and it does not need any changes.

4- Although, the authors have written Results and Discussions very well, the conclusion is not clear and it should be revised and re-organized. What is the future suggestions in this field and what directions should be paid attentions for future researches??

5- All Latin name and scientific names in the main article and even in the Reference list should be Italics. for example in reference 4, Actinidia chinensis is NOT Italics.

6- What is DOI of reference 6???

7- The format of reference 16 is different from other references, please, correct it.

8- If it is possible increase the number of References, especially in discussion part, especially from those published articles in recent years.

Reviewer 2 Report

Dear authors,

In this manuscript, the use of destructive and non-destructive methods to determine the quality of kiwi fruit has been well studied, but it seems that the following points should be taken into account, especially from a statistical point of view, in order to improve the manuscript:

1- The sentence that begins in line 23 seems not to be well and properly connected with the text of the paragraph. Make the necessary change.

2- Write the full specifications of the device (manufacturer and country) (line 114).

3- Were the sensory data normal? They are usually not normal and for this reason, non-parametric methods are used for their analysis. Write enough reasons why you used ANOVA (line 138).

4- The LSD test is suitable for independent comparisons. It is suggested to use another test such as Tukey's test (line 141).

5- It is suggested that the paired observation t-test should be used. Considering that both observations (destructive and non-destructive) are related to the same garden (line 141).

6- Some grammatical adjustments need to be made.

7- Regarding references: reference number 9 was not found in plant science, and references number 15 and 30 were not complete.

Kind regards,

line 37 -indicator and is characterized by the balance

line 72 -During the commercial

line 75 - at Aristotle University

line 188 and 189- orchards

line 194 - exhibited

line 228 - decrease in oxalic acid

line 288- in kiwifruit undergoing ripening

line 308 - acceptance of consumers with respect

line 331 - that received a high

line 339 - resulting in sugar accumulation

line 343 - associated

Reviewer 3 Report

Reading the abstract, the title and the bibliographic references of the article ”Physiological and metabolic traits linked to kiwifruit quality” developed by Vaia Styliani Titeli, Michail Michailidis, Georgia Tanou, Athanassios Molassiotis, we can say that the authors clearly present the purpose of the work, what and how they discovered.

The title of the article is informative and relevant.

Reading the abstract of the article, the reader gets a clear and accurate picture of the research conducted by the authors, being very comprehensive, containing essential introductory information, presenting the used working method and the obtained results.

The introduction clearly presents the topic, clearly outlining the research ideas, presenting in detail what is already known about this topic.

In the material and method chapter, the experience variables are defined and measured properly, the study methods being valid and reliable, there are enough details to repeat the study, the subject of the article being clearly presented.

The research results are presented in an appropriate way, being easy to read and understand, the relevant figures are presented in a clear and visible way. The units of measurement are adequate, the categories of values being grouped accordingly. The text in the results explains the data presented in graphs and tables and does not repeat them, clearly presenting the results obtained. The results are discussed from several angles and placed in context without being overinterpreted.

The conclusions are supported by the obtained results, but also by the bibliographic references, respond to the pursued aim.

Bibliographic references are also relevant and recent, with the authors making a correct reference to generally including appropriate key studies.

The article ”Physiological and metabolic traits linked to kiwifruit quality” developed by Vaia Styliani Titeli, Michail Michailidis, Georgia Tanou, Athanassios Molassiotis, presents an information opportunity for future research, being a consistent article in itself and so this article meets the proposed research objective.
